# Hydrothermal Effect on Mechanical Properties of *Nephila pilipes* Spidroin

**DOI:** 10.3390/polym12051013

**Published:** 2020-04-29

**Authors:** Hsuan-Chen Wu, Aditi Pandey, Liang-Yu Chang, Chieh-Yun Hsu, Thomas Chung-Kuang Yang, I-Min Tso, Hwo-Shuenn Sheu, Jen-Chang Yang

**Affiliations:** 1Department of Biochemical Science and Technology, National Taiwan University, Taipei 11052, Taiwan; 2Graduate Institute of Nanomedicine and Medical Engineering, College of Biomedical Engineering, Taipei Medical University, Taipei 11052, Taiwan; 3Department of Chemical Engineering and Biotechnology, National Taipei University of Technology, Taipei 11052, Taiwan; 4Department of Life Science, Tunghai University, Taichung 400, Taiwan; 5National Synchrotron Radiation Research Center, Hsinchu 300, Taiwan; 6International Ph.D. Program in Biomedical Engineering, College of Biomedical Engineering, Taipei Medical University, Taipei 11052, Taiwan; 7Research Center of Biomedical Device, Taipei Medical University, Taipei 11052, Taiwan; 8Research Center of Digital Oral Science and Technology, Taipei Medical University, Taipei 11052, Taiwan

**Keywords:** spider silk, secondary structures, hydrothermal treatment, strength, *Nephila pilipes*

## Abstract

The superlative mechanical properties of spider silk and its conspicuous variations have instigated significant interest over the past few years. However, current attempts to synthetically spin spider silk fibers often yield an inferior physical performance, owing to the improper molecular interactions of silk proteins. Considering this, herein, a post-treatment process to reorganize molecular structures and improve the physical strength of spider silk is reported. The major ampullate dragline silk from *Nephila pilipes* with a high β-sheet content and an adequate tensile strength was utilized as the study material, while that from *Cyrtophora moluccensis* was regarded as a reference. Our results indicated that the hydrothermal post-treatment (50–70 °C) of natural spider silk could effectively induce the alternation of secondary structures (random coil to β-sheet) and increase the overall tensile strength of the silk. Such advantageous post-treatment strategy when applied to regenerated spider silk also leads to an increment in the strength by ~2.5–3.0 folds, recapitulating ~90% of the strength of native spider silk. Overall, this study provides a facile and effective post-spinning means for enhancing the molecular structures and mechanical properties of as-spun silk threads, both natural and regenerated.

## 1. Introduction

Spider silks are protein fibers consisting of hierarchically synergized protein motifs that account for outstanding mechanical properties (toughest material in nature) and biocompatibility [1,2,3]. The unique motifs, including the polyalanine- and glycine-rich sequences (AAA or GGA), constitute the crystalline and amorphous regions in tandem repeats, imparting the ductility and strength of silk fibers and leading to excellent toughness [4,5]. For instance, the silk fiber produced by the major ampullate gland (MA) of spiders is the strongest amongst various silk types and is utilized to form the radii and frames of spider webs. Such major ampullate silk, also known as the major component of dragline silks, exhibits a tensile strength stronger than steel (>1 GPa), and it also has adequate viscoelasticity (~20–35%) [6]. The incomparable toughness and the excellent biocompatibility of spider silk results in potential applications in industrial and biomedical configurations, such as performance/protective gears, ultra-thin sutures for neurosurgery and wound closure, band aids, and implants for bone, cartilage, tendons and ligaments [7,8].

It is generally recognized that the extraordinary performance of the spider silk resides in the well-organized inter- and intra-molecular chains of silk proteins. Those hierarchically supra-molecular silk structures are greatly controlled by the sophisticated spinning process occurring in the spinning ducts of the spiders. Currently, two common hypotheses describe the hierarchical structures responsible for the spinning process that transforms silk protein solutions into solid filaments at the submicron and micron levels, namely the liquid crystalline model [9] and the micelles model [10,11,12]. The liquid crystalline model describes the transformation of liquid silk protein into fibers formed within the spinning duct of spiders, which in turn is accompanied by increasing shear forces that lead to the formation of intermolecular secondary structures. The micelles model describes the formation of micelle structures through the interplay of the hydrophilic tails and hydrophobic cores of the spidroins and the induction of molecular-chain alignment upon the spinning process [10,11]. Taken together, the spidroin-aligned secondary structures (α-helices, random coil, and β-turns) govern the mechanical properties, e.g., hydrophobic antiparallel β-sheets that provide strength and β-spirals that offer the elasticity of the spider silks [9,11,13,14,15,16,17].

Though the physical properties of spider silk are predetermined by the spinning process of spiders, the molecular interaction within the spun fibers may be subjected to environmental change, leading to a tunable silk performance. This may include a typical phenomenon called the supercontraction of spider silk fibers that involves the shrinkage of spider silk fiber to half of its original length, when the dragline silk is in a hydrated and relaxed state. Those are the silk fibers that possess remarkable alternations in their molecular structures and subsequent mechanical behaviors, owing to the presence of water as a plasticizer, that is used to alter hydrogen bonds [18,19].

Additionally, there exists another crucial factor, which is the thermal history of the molecular structure, as well as the mechanical behavior of spider silk. For instance, there is a heating-induced gradual transition in spider silk structure, attributed to weakened hydrogen bonds and disordered molecular chains in amorphous (at a low temperature range) and β-sheet crystalline regions (at a higher temperature range) [20,21,22]. For mechanical performance, the literature has reported an interesting observation of spider silk, suggesting a unique storage modulus transition at ~60 °C and an elongation at break at ~70 °C, indicating a rise in stiffness or stronger interactions between molecular chains [21,23]. Similar to spider silk, recent studies on silkworm silk have also indicated molecular and physical property transition at ~60 °C due to the enhancement of the dynamic modulus in first temperature scan ranging from 25 to 100 °C [24]. Comprehensively, it is believed that a more precise re-investigation of temperature treatment on spider silk would be worthy in establishing a relationship among the molecular structures, processing, and subsequent mechanical performance.

In this research, a strategy combining hydration and thermal treatments was exerted on spider silk to comprehend the transition of secondary structures and mechanical performance at the microscopic and macroscopic levels, respectively. Using native spider silk as the benchmark material, we could easily evaluate the success of these strategic approaches upon altering the physical characteristics of the treated silk fibers. As a convincing result, a conformational transition of secondary structures (from random coil to β-sheet) and an enhanced mechanical strength after the post-hydrothermal treatment of the native spider silk was witnessed. Furthermore, artificially-regenerated spider silk threads, when treated with the same post-spinning condition, also exhibited altered secondary structures, and an increased mechanical strength. Overall, the hydrothermal treatment on spider silk materials explored in this research serves as an effective means towards reorganizing hierarchy of silk molecules, rendering superior mechanical strength as a favorable outcome. Furthermore, this technique, in turn, could be also scaled-up for the future fabrication of high-performance synthetic silk materials using the recombinant silk sources.

## 2. Materials and Methods

### 2.1. Silk Collection and Preparation of Silk Films

Orb-weaving spiders of the genus *Nephila* are among the most popular species for silk studies. Therefore, for this study, adult female *Nephila pilipes*, weighing from 3.7 to 4.3 g, were chosen to collect silk. Only the dragline produced by major ampullate glands was examined. Forced silking was used to collect dragline silk from *N. pilipes*. Spiders were first placed on a platform, ventral side facing upward and the legs and abdomen fixed with non-sticky tapes and insect pins. Two threads of major ampullate dragline silks were pulled from the spinneret and were taped on a rotor powered by a motor. The platform was placed under a stereo-microscope to ensure that no experimental error or contamination occurred during the forced silking. Dragline silk was drawn at a speed of 2–4 m/min, which was similar to that of natural spinning. Some of the collected silk was prepared for mechanical tests; some was dissolved in HFIP (hexafluoro-isopropanol) (Sigma-Aldrich, Saint louis, MO, USA) at 50 °C for 2 h, dried in aluminum pan at room temperature; and the remaining was cast on glass slides for FT-IR analysis. After the dried films were obtained, both dragline threads and spidroin films were immersed in distilled water in a water bath and heat-treated for 30 min (designated as heat treatment in water) at 30, 40, 50, 70, and 90 °C. The films samples were removed from water bath, immediately dried at ambient temperature, and analyzed by FT-IR. Additional groups of dragline threads and spidroin films were also thermally treated using a heat blower (designated as heat treatment in air), with a tunable temperature controller for 30 min at 30, 40, 50, 70, and 90 °C, followed by a series of FT-IR analyses and tensile tests.

### 2.2. Characterization of Mechanical Properties of Spider Silk Fibers

The strength and elasticity of the collected dragline silk (first subjected to fiber fineness measurement by vibroscope) was measured by a ZWICK 1445 mechanical testing instrument (ZwickRoell, Herefordshire, UK). In each measurement, 20 threads of silk fibers were measured, and the data were averaged to represent the silk properties. In addition to the silk of the *N. pilipes*, the major ampullate dragline silk from the tent-web spider, *Cyrtophora moluccensis,* was harvested for further comparison. Similar forced-silking conditions for collecting the silk of *C. moluccensis*, at 2~4 m/min was utilized, and the strength and elasticity of dragline silk from these two types of spiders were subsequently measured.

### 2.3. Circular Dichroism

The secondary structure conformation of the silk protein in a solution was characterized by circular dichroism (CD). A JASCO spectropolarimeters J-810 (JASCO International Co. Ltd., Tokyo, Japan) at a scan range of 180–260 nm was utilized to examine the spectrum for the spidroin solution. HFIP was chosen as the solvent to dissolve silk protein at the final concentration of 0.5 wt % at 50 °C, 2 h. The CDNN 2.1 software was then used to calculate the secondary structure proportion from the obtained CD spectrum. The structural conformations of the dissolved dragline solution and native liquid gland silk dope from *N. pilipes* were compared. In addition, the dissolved silk from *C. moluccensis* was utilized as a reference for the silk solution of *N. pilipes*.

### 2.4. FT-IR Analysis

The FT-IR spectroscopy was performed for the silk subjected to different heat treatments (30–90 °C). In this experiment, an autoimage attenuated total reflectance Fourier transform infrared spectroscopic (ATR-FT-IR) microscopy system (Perkin Elmer, Llantrisant, UK) was used to probe the silk samples. The resolution of the mercury cadmium telluride (MCT) detector was 4 cm^−1^, and the operation condition of the scan number was 128. After the FT-IR spectra were obtained, software PeakFit^®^ 4.11 (Systat Software) was used for further FT-IR spectrum deconvolution and secondary structure analysis.

### 2.5. Reconstitution of Spider Silk and Microspinning

The 20 g of spider silk samples collected by forced-silking were dissolved in HFIP (boiling point of 58.2 °C) at a final concentration of 10 wt % [25]. The silk protein/HFIP spinning dope was placed in a glass syringe (Hamilton, Model 1002 SL, Bonaduz, Switzerland) equipped with a 26G needle with an inner diameter of 0.5 mm. The syringe with the silk solution was hooked up to a syringe pump (Kd Scientific, KDS-100, Holliston, MA, USA), and the silk protein/HFIP dope was pumped out at flow rate of 10–100 μL/h. Regenerated silk fibers were formed by extruding the silk solution through the syringe needle and subsequently collected on a six-poled winder powered by a customized motor. The collected regenerated-silk samples were secured on the poles of the winder by taping with adhesive tape (water resistant and thermo-stable) and collected with the reeling speed of 60 cm/min. The regenerated fibers, together with the winder, were removed from the motor and submerged into 60 °C double-distilled water (dd H_2_O) for 30 min, followed by immediate drying at ambient temperature. The silk fibers were then removed by carefully cutting them off from poles on the winder, after which, they were evaluated for tensile properties.

### 2.6. Statistical Analysis

For the statistical analysis, experiments were performed in triplicates, the mean ± standard deviation was calculated, and statistical significance was determined by a Student’s *t*-test (*p* < 0.05).

## 3. Results and Discussion

### 3.1. Analysis of Secondary Structures of Silk Fibroin by Circular Dichroism

Spiders have evolved to produce unique/specialized silk proteins and build orb webs (comprised of silk threads) to account for various functional properties and applications [26]. Each of those silk proteins possess diverse material and biological properties that are tailored for specific applications [27,28,29]. In this research, we primarily compared the silk properties between two common spider species, *N*. *pilipes* and *C. moluccensis*, allowing us to screen the ideal silk material in the subsequent experiments of this research. *Nephila* is the largest orb web weaver species, tropical and subtropical region dweller, and are found in Taiwan [22,30]. It makes the vertical orb-web to catch flying insects; while *C. moluccensis*, the largest 3D space-weaver found in Taiwan, builds up the tent webs for the arrest of hopping insects [22].

Specifically, at first, the majority of the ampullate spider silk was collected from the two species of spiders and dissolved separately into HFIP at 0.1 wt %, at 50 °C, for 2 h. This was followed by the measurement of its CD spectra and deconvolution analysis. CD is among the common techniques in studying protein secondary structures, and the deconvolution algorithm provides a relative estimation of secondary structures for proteins in a solution [31]. The secondary structure proportions of *N. pilipes* and *C. moluccensis* silk were calculated and are summarized in Table 1. The data of CD spectra in the solution indicated the similarity of conformation between liquid silk dope and dissolved silk-thread sample of *N. pilipes*. Compared to the secondary structures of silk from *C. moluccensis*, the dragline silk of *N. pilipes* contained more β-sheet structures in proportion, in the form of antiparallel and parallel β-sheets, 28.4% and 8.1 %, respectively. The β-turns were found be similar in liquid and dragline silk from *N. pilipes* in comparison to those from *C. moluccensis*. As previously reported [32,33,34], the β-sheet domain is primarily responsible for the mechanical strength of the silk thread, while β-turns can build up a spiral-like spring structure that could be stretched under extension (renders elasticity). Based on data from Table 1, it is indicative that the dragline threads of *N. pilipes* may exhibit a different mechanical performance in comparison to those from *C. moluccensis*.

### 3.2. Mechanical Strength of Dragline Silk Fibers

It can be seen from Figure 1 that the mechanical strength of dragline silk from *N. pilipes* (1084 ± 355 MPa) was found to be significantly higher than that of *C. moluccensis* (260 ± 91 MPa), while the mean elasticity of the dragline silk from *C. moluccensis* (25.2 ± 8.4%) was barely larger than that from *N. pilipes* (20.2 ± 3.1%), with no significant difference. By combining the results from Table 1 and Figure 1, the enhancement of the mechanical performance for the treated silk samples could possibly be attributed to the elevated β-sheet content in the silk [35]. Moreover, the secondary structure of silk thread was found to be retained, even after re-dissolution, indicating the absence of any ill effects (degradation) of the HFIP treatment on silk [36]. Overall, the dragline silk produced from *N. pilipes* was mechanically superior to that from *C. moluccensis*, and, as such, it was consequently chosen as the study material for the rest of the research.

### 3.3. Effect of Thermal Treatment on Spider Silk Samples

To evaluate the thermal impact of the molecular structures and mechanical performance of spider silk, a series of silk threads drawn from *N. pilipes* were first fixed on the frames, followed by heat treatment (in air or water). Figure 2 represents post-treatment effects at various temperature on the strength of the dragline threads spun by *N. pilipes*. The water treatment was performed by using hot dd H_2_O in a water bath, while for the air treatment, a heat blower with a temperature controller was used as the heat source to treat the silk threads. As depicted in Figure 2, the temperature played a pivot role in varying the strength of dragline threads in both the air and water treatments (ranging from 30 to 90 °C). The heat-treated samples (30 °C in air) were regarded as the reference, indicating the untreated spun silk at the ambient condition. The parabolic curves obtained for the same suggested an enhancement of silk strength with increasing temperature up to the critical point, 70 °C, beyond which, a decrease in the strength was observed. Hence, a temperature in the range of 50–70 °C may be the most effective in promoting the strength of spider silk.

A similar phenomenon on insect silk fibers was reported by Tsukada et al. [23], wherein tussah silk fibers exhibited an enhancement in storage modulus upon increasing the temperature from 30 to 60 °C. This was attributed to a rearrangement of molecules in amorphous regions, leading to strengthening of intermolecular interactions [23]. Additionally, when comparing the two plots from the heat treatment conditions, breaking strength (1202 ± 114 MPa) was higher by 17.18% between 50 and 70 °C in the liquid condition than that in the air condition (1026 ± 99 MPa) (no statistical difference was observed). This indicated a relatively stronger interaction that further occurred during the rehydration of the silk samples, thereby promoting a more complex reorganization of the molecular chains within the spider silk threads [21]. Thus, the effect of hydration and thermal process, taken together as the post-treatment procedure [19], played a favorable role in tuning the structures and properties of the spider silk threads.

### 3.4. FT-IR Analysis of Hydrothermally Treated Silk Samples

To further elucidate the molecular mechanism that the hydrothermal process had exerted upon the spider silk threads, FT-IR analysis was performed. The FT-IR spectra corresponding to each of the spidroin film samples exposed to post-treatment at varied water temperature from 30–90 °C are shown in Figure 3A. The 1600–1700 cm^−1^ region corresponded to an amide I group, which represented the secondary structure conformation [37,38]. The region at 1630–1667 cm^−1^ was responsible for random coils and α-helices [39,40,41], and 1670–1685 cm^−1^ was responsible for turns [39,40,41]. The β-sheet band at 1620–1640 cm^−1^ [39] was prominently increased at 60 °C (represented in red). The major functional group bands/peaks for the β-sheet, α-helix, and random coil were further separated and analyzed by PeakFit 4.11 to demonstrate the occurring structural transition (Figure 3B). The proportion of the β-sheet was calculated as the ratio of area under the peak of the β-sheet to the total of area of the β-sheet, β-turn, α helix, and random coil. An enhancement in the intensity of the β-sheet (41.2%) was observed at 30 °C, which could have been related to the total β-sheet (36.5%), obtained from CD (Table 1). Further, the intensity of β-sheets at 40, 50, 70, and 90 °C were estimated to be 31.7%, 40.1%, 34.5% and 33.0%, respectively. Evidently, at 60 °C treatment, the β-sheet reached the maximum intensity (47.8%), transformed from the random coil conversion, and the β sheet-to-random coil ratio was maximum (three-fold higher), as shown in Figure 3C; a similar trend of this has also been reported [42,43]. Thus, this enhancement in the β-sheet fraction may be regarded as a key index to estimate the strength of structurally stronger silk materials [44].

### 3.5. Mechanical Property of Hydrothermally-Treated Regenerated Spider Silk Fibers

Taking a step further, we evaluated the effect of hydrothermal treatment serving to enhance the tensile strength of the artificially-spun silk fibers. Specifically, (as in Figure 4A), we collected the MA silk out of the *N. pilipes* as the source of our spinning doped material [45]. The silk was subsequently dissolved in HFIP for 2 h at 50 °C [25] and then artificially-spun on the hexagonally-poled winder powered by a customized motor (~60 cm/min spinning rate). The collected silk samples were secured on the poles using adhesive tapes (water resistant and thermally stable). The winder was removed and submerged into a 60 °C dd H_2_O bath for 30 min, followed by immediate drying at ambient temperature. The silk fibers were then carefully removed from the winder and further evaluated for the tensile properties. The final tensile strength of the engineered silk threads was assessed for a further efficacy comparison. As depicted in Figure 4B, the direct micro-spun fiber threads, without any hydrothermal treatment, exhibited an inferior mechanical strength (390 ± 145 MPa) that was about one-third of the native silk fibers (960 ± 141 MPa). However, the micro-spun fibers treated at 60 °C in dd H_2_O exhibited a significantly enhanced strength (910 ± 127 MPa), restoring ~90% of the strength of the native spider silk threads. Such a demonstration revealed a post-processing treatment that offers a favorable outcome towards improving the mechanical performance of the existing fibers. We envision that this could also provide pertinent guidance for current or future silk-based material fabrication and applications that are not limited to the native spider silk demonstrated here.

## 4. Conclusions

In summary, both the molecular structures and physical properties of the major ampullate dragline silks, collected from the *N. pilipes* spider, were characterized. In comparison to that of *C. mollucensis*, the silk from *N. pilipes* exhibited a higher beta-sheet content at the microscopic level, and a stronger tensile strength at the macroscopic performance. A further systematical investigation was performed to reveal the thermal and hydration effects on the silk threads from *N. pilipes*. A configurational transition of secondary structures (random coil to β–sheet) and overall enhanced tensile strength were observed upon the hydrothermal treatment (50–70 °C water) of spider silk. Such an optimal post-treatment processing of natively-drawn silk threads was also applicable to further improve the strength of regenerated spider silk threads (~2.5–3 fold), comparable to the ultimate strength of native spider silks. Building upon such achievements on the natural spider silk, we believe that the proposed hydrothermal treatment could also be of immense interest in fabricating future silk materials of various recombinant sources.

## Figures and Tables

**Figure 1 polymers-12-01013-f001:**
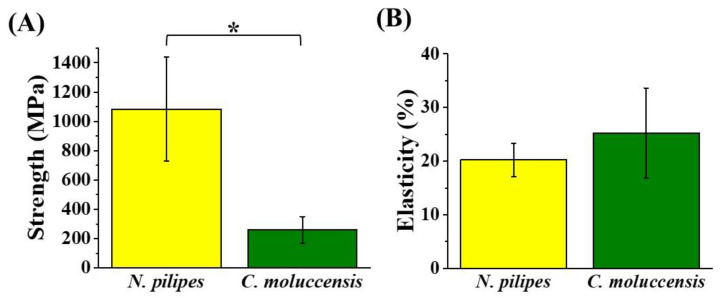
Strength **(A)** and elasticity **(B)** of major ampullate dragline silks collected from *Nephila pilipes* and *Cyrtophora moluccensis* (* denotes *p* value < 0.05 for the strength average in the silk from two spiders).

**Figure 2 polymers-12-01013-f002:**
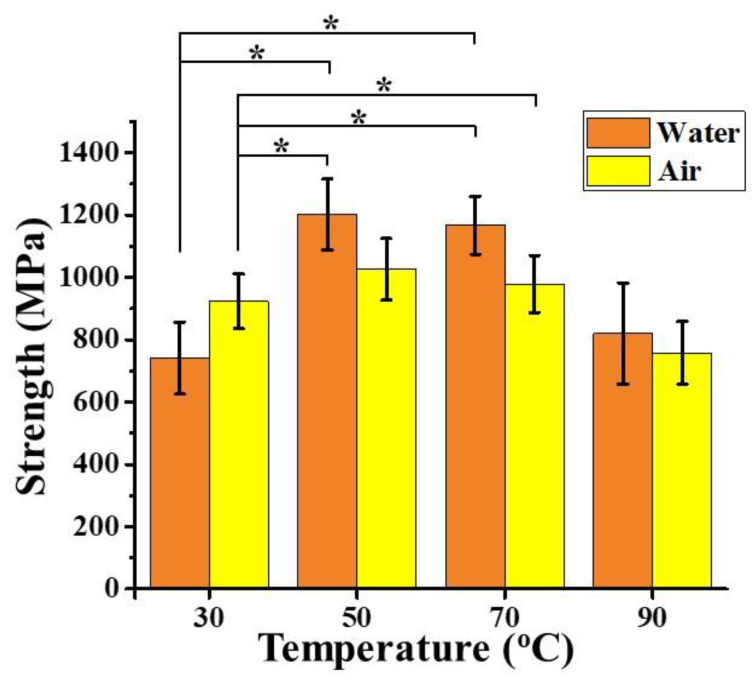
The breaking strength of the hydrothermally-treated dragline silk from *N. pilipes*. The orange columns designate the hydrothermal treatment group (in water), and the yellow columns designate the thermal treatment group (in air). * represents statistically difference (*p* value < 0.05).

**Figure 3 polymers-12-01013-f003:**
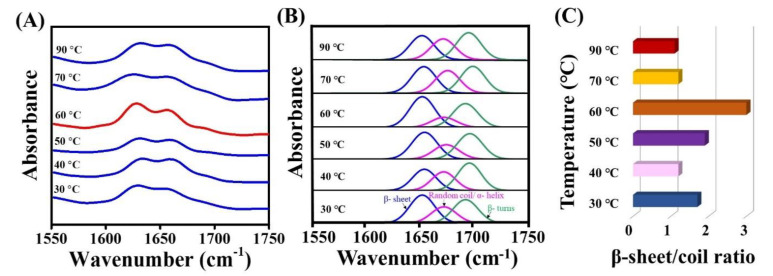
FT-IR structural analysis of spidroin film by hydrothermal post-treatment from 30 to 90 °C. (**A**) FT-IR spectra of treated films. (**B**) Curve deconvolution, peak separation, and assignment of secondary structures from (**A**) deconvoluted blue peak: β-sheet; pink peak: random coils, or α-helices. Green peak: turn structures. (**C**) Estimation of the ratio of β-sheet-to-random coil at various conditions of hydrothermal treatments.

**Figure 4 polymers-12-01013-f004:**
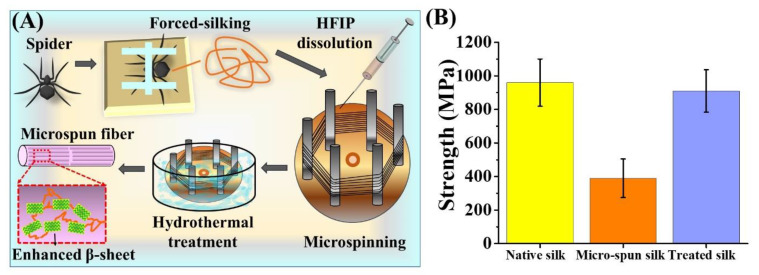
Hydrothermal treatment of artificially-spun spider silk microfibers. (**A**) Schematic representation of the production of regenerated silk fibers. Dragline silks were collected and dissolved into hexafluoro-isopropanol (HFIP), followed by microspinning with a controlled system with the syringe and the hexagonal winder. The fibers were treated with water at 60 °C, which enhanced the β-sheet content. (**B**) The breaking strength of microfibers from the native spider *N. pilipes* subjected to forced-silking, microspinning, and heat-treatments in water.

**Table 1 polymers-12-01013-t001:** Secondary structures estimation of spider silks via CD analysis.

Conformation	Liquid Silk of *N. pilipes* (%)	Dragline Silk of *N. pilipes* (%)	Dragline Silk of *C. moluccensis* (%)
α Helix	19.8	20.7	30.0
Antiparallel β-sheet	27.4	28.4	21.9
Parallel β-sheet	8.6	8.1	7.1
β turn	18.3	18.8	20.3
Random coil	16.0	24.1	20.7

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
