# Peer review of "Hydrothermal Effect on Mechanical Properties of Nephila pilipes Spidroin"

_polymers, 2020, doi:10.3390/polym12051013_

Round 1
Reviewer 1 Report
There are several grammatical errors that need to be corrected but overall it is well written. The data are very interesting and the biophysical analyses are well correlated to the data. It would be nice to see the error bars on Fig. 3C as that is a key element of their discussion for why the improved properties are seen. It is not clear what 3.1 CD analyses means as it was done on dissolved silk in HFIP and that really has little to do with any further data. So that either needs to be left out or related to the key findings of increased tensile strength with hydrothermal treatments. There is also not data on the effects of the treatments on elongation/elasticity. Also representative stress/strain curves would be nice to see to compare to the normal silk.
Reviewer 2 Report
The manuscript entitled "Hydrothermal Effect on Mechanical Properties of Nephila pilipes Spidroin" attempts to define the structural reorganization that occurs when both native and reconstituted major ampullate silk are heated. There are significant problems in the text both in how the authors utilize the English language and also in how they describe their material/ methods and results.
Abstract: There are numerous problems throughout the abstract that require extensive editing. Further, the authors state "....provides an ideal model towards reprocessing the as-spun silk thread for enhanced molecular structures and mechanical properties." I would argue that while the premise is correct, reprocessing silks is not viable. Spiders cannot be farmed due to their territorial nature and thus, collecting silk and reprocessing it is of little to no value. There is however value in how synthetic spider silks are processed and this is not mentioned anywhere in the manuscript.
Examples:
Line 52: ....enormous industrial and biomedical potentials,... hyperbole.
Line 55: "While the secrets of extraordinary performance residing in the spider silks somehow remain to be explored completely, and those are believed to be strongly-associated with the spinning processes of the spiders." This sentence needs to be revised as it does not make sense.
Line 59: Presenting two models for protein storage and missing a key citation. Parent et al () clearly present results that the micellar confirmation is supported with experimental evidence.
Line 90: Further example of sentence structure/language "Furthermore, same post-treatment condition exerted on the artificially regenerated spider silk threads also increased the mechanical strength significantly."
Line 101: "Two threads of major ampullate (from the cut opened gland) silks were pulled from the spinneret and were taped on a rotor powered by a motor." From reading the results, the glands were never cut open. Further, forced silking is not done by dissecting the spider. This clearly needs revision and clarification.
Line 107: "...at 50 °C for 2 h, dried in aluminum pan, at room temperature, and used for circular dichroism..." This does not appear in line with what is reported in the results. CD was performed on HFIP solubilized silks and not on silk films.
Line 108: "After the dried films were obtained, both dragline thread and spidroin film were immersed in distilled water in a water bath and heat-treated for 30 minutes, at 30 °C, 40 °C, 50 °C, 60 °C, 70 °C, and 90 °C." This reads as if the only treatment that they performed was a water bath based heat treatment. In the results, the authors report water bath and dry heat treatment.
Line 136: The method of microspinning is insufficiently described. See the general comment below.
Line 159: Circular Dichroism is described as a "power tool".
Line 178: "...the outcome unveiled the role of β-sheet in enhancing the mechanical strength of spider silk fibers." The role of the beta-sheets in mechanical properties has been well studied and reported.
Line 192: Authors describe the "heat blower" treatment from the previous comment regarding line 108. This is inadequately described in the results section.
Line 245: Again, insufficient description of the spinning technique without reference.
A more general comment; It has been widely established that stretching of spider silks, native and recombinant, forces beta sheet alignment along the axis of stretch (the fiber). That is not discussed. Given that the spinning method is insufficiently defined, it is impossible to ascertain if stretch was maintained between sample batches or if it varied. If it was varied, that could very easily account for the differences in mechanical properties between the treatment groups.
Reviewer 3 Report
Hydro-thermal treatment of spider silk is presented with improvement of its mechancal properties. It is interesting study with combination of natural and artificially produced silk. Several techniques were combined for characterisation.
Several issues should be resolved:
- Please show CD spectra and which wavelength corresponds to which type of the structure. This is very useful information to make comparison with moth silk: Silk: Optical Properties over 12.6 Octaves THz-IR-Visible-UV Range, A Balčytis, M Ryu, X Wang, F Novelli, G Seniutinas, S Du, X Wang, J Li, ...Materials 10 (4), 356 2017
- It would be useful to present analysis at IR spectral range where orientation and silk structure is usually examined:
Appl. Sci. 2019, 9(19), 3991; https://doi.org/10.3390/app9193991
Orientational mapping augmented sub-wavelength hyper-spectral imaging of silk
M Ryu, A Balčytis, X Wang, J Vongsvivut, Y Hikima, J Li, MJ Tobin, ...
Scientific Reports 7 (1), 7419 2017Nanoscale optical and structural characterisation of silk
3. In moth silk, hydro-thermal treatment is usually carried in ethanol-methanol solutions at comparable temperatures as in the reported experiments. If only water is used, silk fibroin is not making hydrogen bonding into beta-sheets. Also other thermal and ionisaition methods can be used: Silk fibroin as a water-soluble bio-resist and its thermal properties
M Ryu, R Honda, A Cernescu, A Vailionis, A Balčytis, J Vongsvivut, JL Li, ...
Beilstein journal of nanotechnology 10 (1), 922-929 2019
J Morikawa, M Ryu, K Maximova, A Balčytis, G Seniutinas, L Fan, ...
RSC Advances 6 (14), 11863-11869 2016Silk patterns made by direct femtosecond laser writing
K Maximova, X Wang, A Balčytis, L Fan, J Li, S Juodkazis
Biomicrofluidics 10 (5), 054101 2016
Why pure water treatment was chosen, even when it is known to be less efficient?
Please show the image of fabricated artifical silk. When mechanical properties are compared, please, indicate the diameters of the fibers; natural and artificial. Proper scaling needs those parameters.
Have the artificial silk fibers acquired optical birefringence at visible wavelengths as a consequence of beta-sheet formation?
Round 2
Reviewer 2 Report
The authors of "Hydrothermal Effect on Mechanical Properties of Nephila pilipes Spidroin" have made an effort to address the previous review. However, the manuscript continues to need editing and continues to suffer from general confusion regarding methods and results. Some major points of concern are listed below.
This reviewers previous concern about an inadequate description and control of the spinning method for their reconstituted silk remains. While the authors focused on post-spin treatments, the spinning process is critically important to control to ensure that any effect observed is known to only be the result of the post-treatment and not the spinning process. Again, there are critical aspects missing from the revised description. The fibers were spun through the air but the description of how that occurred is lacking. Was a droplet formed at the end of the syringe needle and a fiber drawn from that? If so, how was the rapid drying of HFIP controlled for from the droplet and how does the exterior drying of the droplet effect fiber formation and the force required (stretch) to pull a fiber from that drying droplet?
Continuing on, the author's own data indicates that 60C is the ideal treatment temperature due to increased beta-sheet content. However, when one looks for the mechanical data that should support improved mechanical characteristics, it is missing. The authors report 30C, 50C, 70C, and 90C. If 60C is the best treatment group (according to increased beta-sheet content) the mechanical data should be presented.
Also, the authors have chosen not to report any mechanical data beyond tensile strength. Elasticity and youngs modulus are also critically important aspects to consider. Converting random coil and helices to beta-sheets is known to improve the tensile strength but that does not support the assertion that the fibers have regained materials properties approaching that of native spider silk. Spider silk is renowned for its combination of tensile strength and elasticity as that gives it high energy to break.
Further confusion can be noted in Figure 1. Circular dichroism, as described by the authors, is done on dilute solutions of dragline silk fiber in HFIP. That begs the question, what is "liquid silk dope"? No method of collection is described in the materials and methods. Was this material collected from the gland of the spider and then diluted in HFIP as one would surmise?
A final critique, structural analysis was performed on films that were treated in a similar fashion to the reconstituted fibers. This is presented as a direct comparison when it is not. Again, there is some stretch involved in the fiber spinning process as reported. However, the films are not stretched at all. Rather, they are cast as films, allowed to dry and then subjected to FTIR. Without stretching the films, this is not a direct comparison. This reviewer understands the limitations of material when silking spiders. However, if there was enough material to form films there was also enough material to spin it into fibers and use FTIR to study its protein structures to make the requisite direct comparison. This also underscores the need to understand the spinning process to a much higher degree so a reader or reviewer can understand the forces placed upon the fiber prior to post-treatment.
Reviewer 3 Report
remarks have been answered
Author Response
Very appreciate for your kindness.